# Social Network and Bibliometric Analysis of Unmanned Aerial Vehicle Remote Sensing Applications from 2010 to 2021

**Jingrui Wang, Shuqing Wang \*, Dongxiao Zou, Huimin Chen, Run Zhong, Hanliang Li, Wei Zhou and Kai Yan** 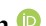

School of Land Science and Techniques, China University of Geosciences, Beijing 100083, China; 2111190012@cugb.edu.cn (J.W.); dongxiaozou@cugb.edu.cn (D.Z.); 2012200022@cugb.edu.cn (H.C.); 2104200077@cugb.edu.cn (R.Z.); lihanliang1998@cugb.edu.cn (H.L.); zhouw@cugb.edu.cn (W.Z.); kaiyan@cugb.edu.cn (K.Y.)
\* Correspondence: wangsq@cugb.edu.cn

**Abstract:** Unmanned Aerial Vehicle (UAV) Remote sensing (RS) has unique advantages over traditional satellite RS, including convenience, high resolution, affordability and fast acquisition speed, making it widely used in many fields. To provide an overview of the development of UAV RS applications during the past decade, we screened related publications from the Web of Science core database from 2010 to 2021, built co-author networks, a discipline interaction network, a keywords timeline view, a co-citation cluster, and detected burst citations using bibliometrics and social network analysis. Our results show that: (1) The number of UAV RS publications had an increasing trend, with explosive growth in the past five years. The number of papers published by China and the United States (US) is far ahead in this field; (2) The US has currently the greatest influence in this field through the largest number of international cooperations. Cooperation is mainly concentrated in countries and institutions with a large number of publications but is not widely distributed. (3) The application of UAV RS involves multiple interdisciplinary subjects, among which "Environmental Science and Ecology" ranks first; (4) Future research trends of UAV RS are expected to be related to artificial intelligence (e.g., artificial neural networks-based research). This paper provides a scientific basis and guidance for future developments of UAV RS applications, which can help the research community to better grasp the developments of this field.

**Keywords:** Unmanned Aerial Vehicle (UAV); Remote Sensing (RS); Bibliometric; Scientometric; visualization

## 1. Introduction

Rapidly-developing Unmanned Aerial Vehicles (UAVs), which stand out with high spatial resolution, short revisit periods as well as lower operating costs, are more suitable for real-time monitoring applications [1]. Given these advantages, applications based on Unmanned Aerial Vehicle (UAV) Remote sensing (RS) have been increasing under the strong investments and support of various governments. The past decade witnessed an expansion of UAV RS applications, from military to civilian uses [2,3]. For example, this technology was used to assess the spatial variability of water conditions in vineyards [4]; to accurately distinguish urban vegetated areas [5]; to map water stress of ground samples and peach orchards [6]; and to determine the coal-burning range of coal mines [7].

To better use UAV RS technology, it is necessary to appreciate the key advances of this technology and how they have evolved. Traditional review papers focus on the content of publications, while the bibliometric method provides a novel solution which pays attention to the characteristics of the countries, institutions, disciplines, and how the papers in this emerging field have evolved. Bibliometrics originated from the early 20th century and has developed into a visual analysis method based on co-citation networks [8]. The CiteSpace software is an information visualization tool developed by Chen [9,10] based on scientific methods (e.g., Social Network Analysis (SNA), clustering analysis, etc.) and

map drawing functions. Data visualization not only identifies and analyzes information through relational data, but also sees the structural relationship among data in a more intuitive way, leading to more in-depth analyses of information [11].

With the technological development of UAVs and the continuous expansion of their applications in various fields, understanding their development has become indispensable. In this study, we used bibliometric analysis and SNA to mine the literature to reveal the hidden structure of this research domain [12], and used visualization methods to quantitatively reflect co-authorship and subject distribution in the UAV RS research field. We identified cutting-edge dynamics and development trends through algorithmic operations [9,13]. This paper explored the UAV RS development history, the annual number of publications, the co-authoring status of countries and institutions, the disciplines involved in this field, and current research hotspots. Through this systematic review of the UAV RS, we aim to offer a better understanding of the overall development of this field and provide a scientific basis for further UAV RS development.

## 2. Materials and Methods

The literature data of our review was derived from the Web of Science (WoS) core database. The search formula was "(TS = ("unmanned aerial vehicle" OR UAV OR "Low-altitude aircraft" OR "Drone")) AND (TS = ("remote sensing" OR RS)) AND Languages: (English) AND Types: (Article OR Review)", which generated 1968 records between 1 January 2010 to 16 April 2021. Since our article only focuses on the application and development of UAV RS, we manually conducted data cleaning to filter unmatched papers. Finally, 1812 articles or reviews were retained. The publication data contained information such as publication date, authors, journal name, country, research institution, keywords, etc. When we analyzed the annual trend of the publications, data for the year 2021 (233 papers) were excluded because their records were incomplete.

CiteSpace can be used quickly and intuitively to reveal the distribution and rules of data. First, we used CiteSpace (5.7.R2) to de-duplicate the retrieved literature data to obtain 1812 articles or reviews, and then built the database. Then, we extracted the corresponding field information through a series of algorithmic operations and drew the scientific collaboration networks of countries and institutions and the discipline interactions networks. In addition, we conducted hotspots analysis by extracting the keywords from the papers and constructing the co-citation cluster map. With the help of CiteSpace, we visualized and analyzed the data rules to show the development of UAV RS applications in a systematic way. The specific research methods were as follows.

### 2.1. Publication Outputs

Firstly, we extracted each country's data from CiteSpace to map the global geographic distribution of the publications and better understand the global development status. Taiwan, Hong Kong, and Macau were merged into China. England, Wales, Northern Ireland, and Scotland were merged into the United Kingdom. The total publications by country were rendered in ArcMap (version 10.5).

Secondly, setting one year as an interval, we drew a line chart of the total publications and a yearly publications bar chart of the top six countries from 2010 to 2020, with the purpose to identify the trend in the number of publications in each country.

Thirdly, we counted publications of institutions over the world with CiteSpace to obtain the publications by institutions within each country.

For publications with one author, the institution and country to which the author belongs increased by one count. For papers that had contributions from multiple authors belonging to different institutions or countries, the number of publications by the corresponding institutions and countries increased respectively.

*2.2. Correlation Characteristics*

We drew the cooperation network and discipline network by establishing a social network analysis (SNA) [14]. SNA not only is a quantitative method that considers the individuals interdependence, but also is a type of structural analysis that can visually display the overall network structure, the position of an individual in the network, and the relationship with other individuals [12,14,15]. Density can be used to measure the integrity and complexity in the network representing the number of connections between points [16]. It is defined as the actual connections number divided by the theoretical maximum connections number, which can be described as follows [16]:

$$\text{Density} = \frac{2m}{n(n-1)} \tag{1}$$

where *m* represents the total of connections, and *n* represents the nodes in the network.

Each node in the network represents one sample (e.g., for a country network, each node represents one country). A node size and its color correspond to co-citation frequency and the first occurrence time, respectively, and the change in color from blue to yellow indicates the time in chronological order. The connection between nodes indicates the co-citation relationship and its thickness indicates the co-citation strength. One node centrality indicates the connection strength with other nodes, and the node with a thicker purple ring represents a higher centrality value and more connections with other nodes. The mathematical definition of centrality is as follows [17]:

$$\text{Centrality}\ (node_i) = \sum_{i \neq j \neq k} \frac{\rho_{jk}(i)}{\rho_{jk}} \tag{2}$$

where $\rho_{jk}$ represents the number of shortest paths between node *j* and node *k*, and $\rho_{jk}(i)$ is the number of those paths that pass through node *i*.

*2.3. Research Front*

Keywords are words and phrases that summarize the research theme of a paper. In our study, keywords were also used to infer the research trends and hotspot changes in the UAV RS field [18]. Using a time slicing technique to build multiple time network models over time, we synthesized these individual networks to form a timeline map [13]. Co-citation analysis showed the clusters that are mainly concentrated in the research field and indicated the current research status. Detecting explosive references can quickly highlight the latest research directions in the UAV RS field.

To summarize the research hotspots and development trends of the UAV RS applications, we selected the top 50 cited papers in each year during the period from 2010 to 2021, constructed a keyword timeline view, counted the top three keywords (except UAV, RS) of new appearances each year, built a co-citation cluster map, and detected the top 25 publications with burst citations.

**3. Results**

*3.1. Analysis of Publication Outputs*

The top ten publishing countries, according to our data in Figure 1, were China, the United States (US), the United Kingdom (UK), Spain, Italy, Germany, Australia, Canada, France and Brazil. The top three countries accounted for 56.13% of the total publications. Only China and the US published more than 300 papers, with 373 and 514 publications, respectively. The US published 243 papers more than the UK. In addition, China and the US accounted for 28.37% and 20.58% of the total publications, respectively, far exceeding the total of the other eight countries.

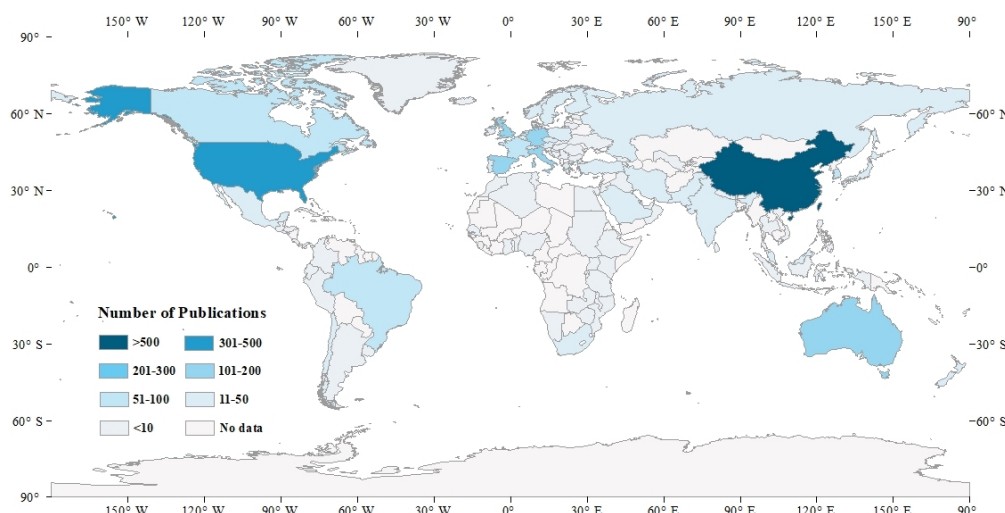

**Figure 1.** The number of global geographic distribution publications from 2010 to 2021.

Publications on UAV RS have shown rapid growth trends between 2010 and 2020 in Figure 2. From 2010 to 2014, a total of 91 papers were published (5.76% of all publications). Publications increased slowly during this period with an average of 18 papers per year. From 2015 onwards, we found 1488 publications (94.24% of all publications), suggesting that the field grew rapidly after this year, with an average annual number of publications of 248. The use of UAV RS has increased widely with the popularization of UAV RS technology and methods, receiving increasing attention from the community [19]. The top ten publishing institutions (Table 1) published 19.32% of the papers. Among them, Chinese institutions occupied an important position (73.14% of the total publication of the top ten institutions). The top three publications were all from China (CAS, WHU, and UCAS).

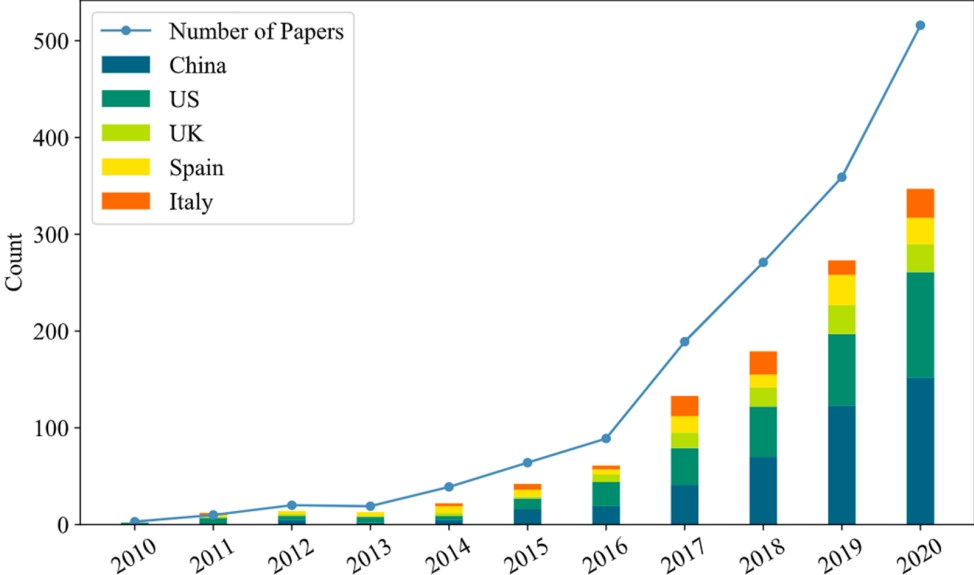

**Figure 2.** The annual publications of countries and the publication trend in the field of UAV RS from 2010 to 2020.

**Table 1.** Top ten institutions publishing papers on UAV RS.

| Rank | Institution | Count |
|---|---|---|
| 1 | Chinese Academy of Sciences (CAS) | 94 |
| 2 | Wuhan University (WHU) | 50 |
| 3 | University of Chinese Academy of Sciences (UCAS) | 44 |
| 4 | Consejo Superior de Investigaciones Cientificas (CSIC) | 33 |
| 5 | Beijing Normal University | 27 |
| 6 | United States Department of Agriculture (USDA) Agricultural Research Service (ARS) | 25 |
| 7 | South China Agricultural University | 23 |
| 8 | Zhejiang University | 18 |
| 9 | University of Twente | 18 |
| 10 | University of Florida | 18 |

### *3.2. Analysis of Correlation Characteristic*

### 3.2.1. Cooperation Network Analysis

The coauthor network reveals the social connections in the scientific community of the research field. The density of the country network was 0.1288, with 703 cooperative relationship pairs among 105 countries, and that of the institution network was $7.5 \times 10^{-3}$, with 547 cooperative relationship pairs among 382 institutions. This indicated that the relationship among the countries was closer than the relationship among the institutions. However, in general, the relationship strength of the above two networks were weak.

In the country network (Figure 3), China ranked first in publications with a centrality of 0.09, and the US ranked second in publications with a centrality of 0.31. However, the US's centrality was three times greater than China's. This indicated the US not only was the most active country in the UAV RS field but also had more communications around the world and showed a wide range of influence. South Korea ranked 12th in the total publications with a centrality value of zero, which means that most of its research was done independently and there was little communication with the outside world. From the network of institutions (Figure 4), we can see that CAS, with 94 papers and a centrality value of 0.23, was the top one in the number of institution publications and had the highest centrality status. WHU ranked second with 50 papers, which was twice that of the USDA ARS publications, but the centrality of USDA ARS was 0.12, which was more than twice that of WHU's centrality. Only the centralities of CAS and USDA were greater than 0.1. Overall, we can infer that cooperation in the UAV RS field was concentrated in countries and institutions with a large number of publications, showing that cooperation was concentrated rather than widely distributed.

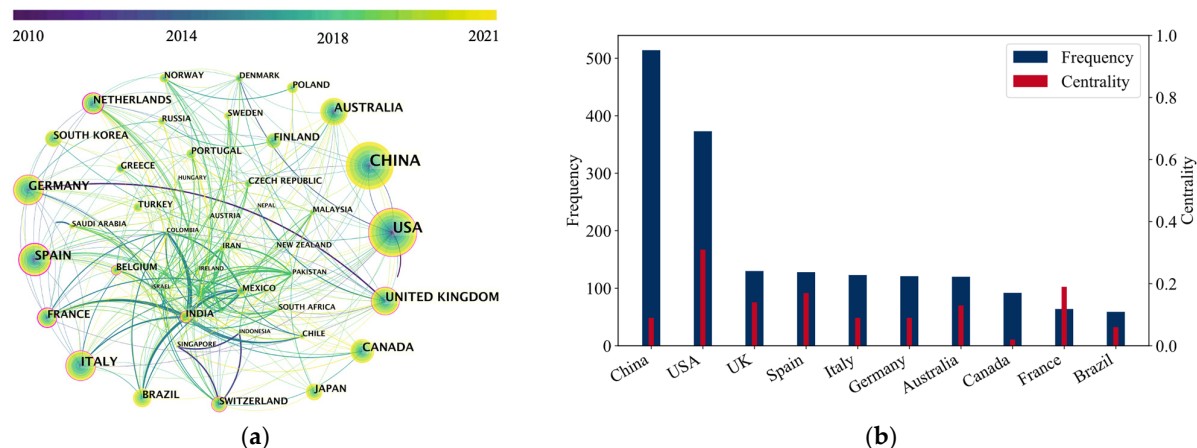

(**a**)  (**b**)

**Figure 3.** (**a**) Collaboration networks of countries; (**b**) The centrality and frequency of the top ten countries.

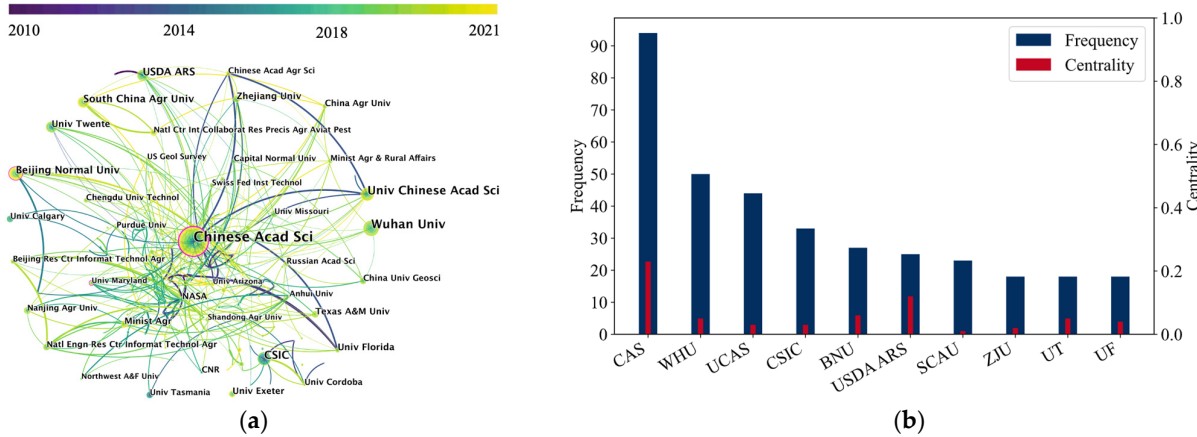

**Figure 4.** (**a**) Collaboration networks of institutions; (**b**) The centrality and frequency of the top ten institutions.

### 3.2.2. Discipline Interaction Analysis

To clarify the disciplines involved in the UAV RS field, we drew the discipline interaction networks in Figure 5. The density of the map was 0.0611, indicating that there was a weak cooperation among disciplines. The network was mainly based on "Environmental Sciences and Ecology" as the main related disciplines, with 893 relevant papers and a centrality value of 0.64. Secondly, UAV RS was widely used in "Photographic Technology", "Geosciences", "Agriculture", "Engineering and Electrical", "Computer Science", "Physical Geography", "Water Resources", among other subjects. In the case of UAV RS used for low-altitude crop photography, crop growth monitoring, and water environmental monitoring were widely used; the entire research process involves the intersection of multiple disciplines, from agronomy to water resources, environmental science, engineering, and ecology. At the same time, when UAV RS is used in land resource surveys, three-dimensional reality simulations, urban planning and other fields, computer platforms are also involved. The processing of images taken by UAVs promotes the development of computer science and electronics. The UAV RS field can be applied to a wide range of topics, and research related to UAV RS is also continuously increasing so that breakthroughs can be found from existing research.

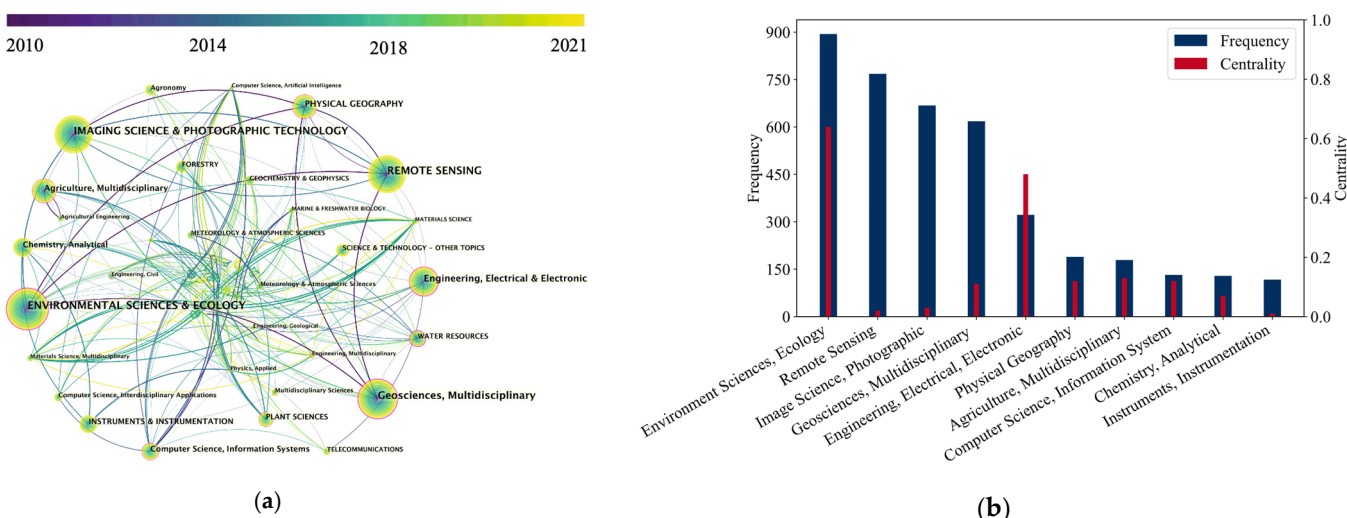

**Figure 5.** (**a**) Discipline interaction networks; (**b**) The centrality and frequency of the top ten disciplines.

### 3.3. Analysis of Research Front

3.3.1. Keyword Analysis

The timeline graph focuses on the relationships among clusters and the historical span of papers within a certain cluster. The nodes of a given cluster are arranged on the same horizontal line in chronological order. The papers in each cluster are like strings on a timeline, showing the clusters' historical results [9]. For example, in Figure 6 the evolution process of the keyword node content in the cluster "#0 vegetation index" from 2010 to 2021 included "ndvi", "vegetation index", "hyperspectral", "crop", "biomass", "precision agriculture", "chlorophyll content", "leaf area index", "winter wheat", "maize", and "plant height". Among them, the "vegetation index" in 2010 was the largest node. UAV RS images were used to calculate vegetation index in the early days. With the development of UAV RS, it was equipped with higher resolution hyperspectral sensors to monitor farmland soil characteristics and, crop growth. Meanwhile, the vegetation index was used to estimate biophysical parameters to produce water stress detection images of leaf area index, chlorophyll content, photochemical reflectance index, and canopy temperature to guide production and supervision [20,21]. Therefore, "vegetation index" was the basics keyword in this cluster, and other keywords were extended and developed on this node.

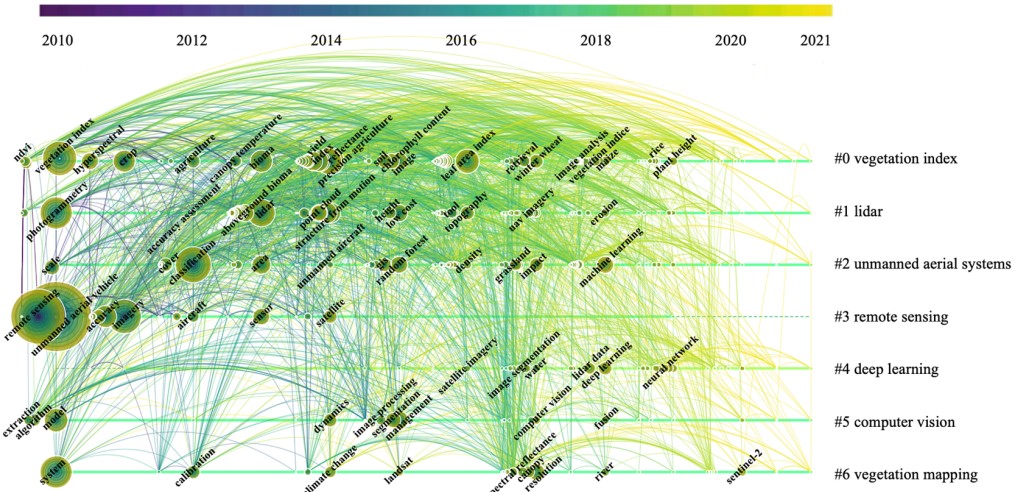

**Figure 6.** A timeline view of keywords in the UAV RS field from 2010 to 2021. Each cluster is shown from left to right, and the view top shows the release time legend. The clusters are arranged vertically in descending order of their size on the right. The colored curve represents keywords appearing in the same paper in the corresponding color year.

From 2010 to 2013, the hotspots in Table 2 were mainly "vegetation index" "photogrammetry", "system", "accuracy", "lidar", "imagery", "classification", "reflectance", etc. The clusters were mainly concentrated in "#0 vegetation indices", "#1 lidar", and "#6 vegetation mapping", involving RS of geology, agricultural science and ecology, and technologies such as image classification and 3D modeling. These hotspots and clusters reflect the more recent period of research hotspots, in which researchers are taking advantage of UAV RS features such as its high resolution, low cost, and high timeliness. Therefore, simple UAVs equipped with sensors are gradually used to monitor farmland soil properties, crop growth and other information in real-time [22]. The high-resolution images provided by UAV RS have improved the accuracy of land vegetation classifications [20]. However, the image classification accuracy is affected by complex surface information, data preprocessing and selection methods. The RS images classification algorithms range from supervised and unsupervised classification to a variety of advanced algorithms, such as support vector machines, neural networks, expert systems, etc. [23]. Multiple classification methods combinations can improve accuracy. Future research directions can start from the perspective of how to integrate multiple classifiers for specific problems [5]. Lidar measures

the distance, surface characteristics and orientation of the target by actively emitting laser light. It can obtain high-precision three-dimensional point cloud data, which is helpful for the rapid establishment of three-dimensional models [24]. With the help of the UAV platform, lidar can be applied to engineering surveying and deformation monitoring [25], disaster assessment [26], forestry surveying [27], forest resource survey [28], etc.

**Table 2.** Top three keywords that newly appeared every year in the period 2010–2013.

| Year | Keyword | Count |
|------|---------|-------|
|      | Vegetation index | 263 |
| 2010 | Photogrammetry | 183 |
|      | System | 163 |
|      | Imagery | 160 |
| 2011 | Precision agriculture | 104 |
|      | Accuracy | 69 |
|      | Classification | 205 |
| 2012 | Lidar | 114 |
|      | Reflectance | 100 |
|      | Bioma | 92 |
| 2013 | Modal | 113 |
|      | Crop | 67 |

In the period 2014–2017, research hotspots included "structure from motion", "forest", "variability", "leaf area index", "chlorophyll content", "resolution", "canopy", etc (Table 3). It can be seen in the timeline view that the clusters were "#0 vegetation indices", and "#1 lidar". With the development of 3D modeling, the effects that can be achieved can no longer meet the needs of researchers, so there is an urgent need to develop key technologies to improve modeling capabilities. Among them, motion recovery structure is a key technology in computer vision 3D reconstruction. The ability to extract 3D point clouds can be improved with the help of UAV systems, and the occurrence of deviations and occlusions can be reduced [29]. In addition, UAV RS involves ecological diversity and ecological restoration and is used to assess forest restoration and habitat quality [30], using aerial images of canopy gaps to assess the understory vegetation plant diversity, which expands forest dynamics and research forest restoration scope [31]. UAV RS system can effectively use a vegetation index to estimate biophysical parameters [21] to produce water stress detection images of leaf area index, chlorophyll content, photochemical reflectance index and canopy temperature to guide production and supervision [32]. Multi-rotor UAV platforms are equipped with multispectral sensors and high-definition digital cameras to form a UAV agricultural monitoring system, which can effectively invert soybean leaf area index and provide new approaches for precision agriculture [33].

**Table 3.** Top three keywords newly appeared every year during the period 2014–2017.

| Year | Keyword | Count |
|------|---------|-------|
|      | Structure from motion | 76 |
| 2014 | Forest | 69 |
|      | Variability | 47 |
|      | Random forest | 46 |
| 2015 | Chlorophyll content | 37 |
|      | Low cost | 35 |
|      | Leaf area index | 89 |
| 2016 | Topography | 28 |
|      | Nitrogen | 18 |
|      | Winter wheat | 33 |
| 2017 | Resolution | 31 |
|      | Canopy | 24 |

The research hotspots during the period 2018–2021 (Table 4) were "machine learning", "deep learning", "maize", "plant height", "neural network", and the clusters were "#4 deep learning", and "#5 computer vision". The stage center has evolved from simple crop estimation in the region to deeper research. For example, UAV RS technology provides a simple, fast, and effective method for small-scale maize lodging investigations. Maize canopy height can accurately identify corn lodging when combined with color features information [28]. Oblique photogrammetry can improve work efficiency by effectively reducing the distortion in the model caused by the lack of ground control points [34]. The multi-spectral camera carried by eight-rotor UAVs can be used to monitor a test area, and the leaf area of cereal crops can be monitored with UAV RS to provide yield estimation models that can quickly and effectively evaluate growth and yield [35]. Bamboo forests image data are accessed by UAV, and then the spectral characteristic information and texture characteristic information of the image are analyzed by using object-oriented multi-scale segmentation. The K-Nearest Neighbor algorithm is used to classify images based on the characteristic band combinations and used to identify different bamboo stands heights [36]. UAVs are expected to be further developed in terms of intelligence, automation, miniaturization and integration. RS image classification and recognition were first used to distinguish the features through manual visual interpretation. This method not only requires specialized staff but also takes a long time. With the development of computer vision technology, image features were gradually applied to RS image classification and recognition, but this method requires a large number of training samples and expert knowledge, which is often difficult to satisfy in practice. As the amount of RS data increases, relying on traditional methods for interpretation and analysis can no longer meet researchers' and operational needs. In recent years, the rapid development of deep learning methods in the computer vision field provides new approaches for RS image classification, target recognition, image segmentation and other fields [37]. For example, research on UAV RS image processing and the application of convolutional neural networks in UAV RS image classification and recognition, and classification and recognition of different types of vehicles [38].

**Table 4.** Top three keywords newly appeared every year in 2018–2021.

| Year | Keyword | Count |
|------|---------|-------|
| 2018 | Machine learning | 63 |
|      | Maize | 21 |
|      | Multispectral imagery | 8 |
| 2019 | Impact | 33 |
|      | Plant height | 12 |
|      | Productivity | 8 |
| 2020 | Deep learning | 40 |
|      | Neural network | 16 |
|      | Damage detection | 16 |
| 2021 | Feature extraction | 8 |
|      | Change detection | 5 |
|      | Rainfall | 4 |

### 3.3.2. Co-Citation Analysis
Cluster Analysis

Different colored areas indicate when the citation links in these areas first appeared. The purple region had the earliest appearance, followed by the green area and later, in order, by the blue and the yellow areas. Each cluster can be labeled with title terms, keywords, and abstract terms that refer to papers in the cluster. For example, in Figure 7 the green area in the lower corner was labeled "#6 using deep learning", indicating that papers about deep learning refers to cluster #6. There were 11 clearly identifiable clusters in the co-citation network. The largest node in the cluster graph was a publication written by Colomina I,

which reviewed the development of UAV RS. The other nodes with red rings were papers with burst references.

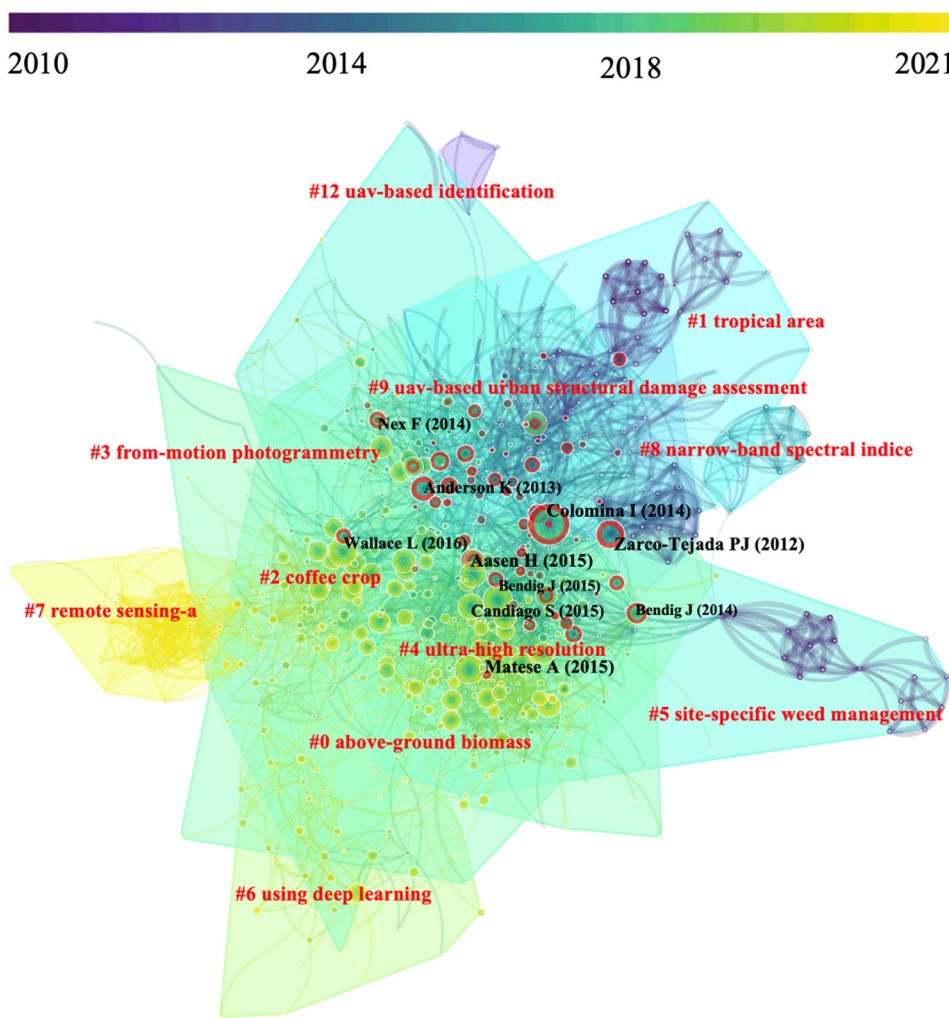

**Figure 7.** A landscape view of the co-citation network in the UAV RS field. The red circles represent the publications with burst references. The top ten publication authors cited in the map are marked (among those cited in citation bursts).

Most of the research in the top ten cited publications (Table 5) were related to forestry and agronomy, which was consistent with vegetation index ranking first in the keyword analysis between 2010 and 2021. Related clusters in the co-citation network included "#0 above-ground biomass", "#2 coffee crop", "#3 from-motion photogrammetry" "#4 ultra-high resolution", "#8 narrow-band spectral indices", and "#12 uav-based identification". Large nodes or nodes with red tree rings were of particular interest, especially the top ten cited publications. Bendig J used vegetation index and plant height information obtained from UAV images to monitor the biomass in barley [39]. Matese A compared the precision viticulture on multiple RS platforms [40]. Aasen H and Wallace L studied the application of UAV in vegetation monitoring and forest structure assessment, respectively [41,42].

**Table 5.** The top ten cited publications in the UAV RS field.

| Title | Author | Journal | Year | Frequency |
|---|---|---|---|---|
| Unmanned aerial systems for photogrammetry and remote sensing: A review [29] | Colomina I | ISPRS Journal of Photogrammetry and Remote Sensing | 2014 | 155 |
| Combining UAV-based plant height from crop surface models, visible, and near infrared vegetation indices for biomass monitoring in barley [39] | Bendig J | International Journal of Applied Earth Observation and Geoinformation | 2015 | 82 |
| Intercomparison of UAV, Aircraft and Satellite Remote Sensing Platforms for Precision Viticulture [40] | Matese A | Remote Sensing | 2015 | 78 |
| Generating 3D hyperspectral information with lightweight UAV snapshot cameras for vegetation monitoring: From camera calibration to quality assurance [41] | Aasen H | ISPRS Journal of Photogrammetry and Remote Sensing | 2015 | 68 |
| Assessment of Forest Structure Using Two UAV Techniques: A Comparison of Airborne Laser Scanning and Structure from Motion (SfM) Point Clouds [42] | Wallace L | Forests | 2016 | 66 |
| UAV for 3D mapping applications: a review [43] | Nex F | Applied Geomatics | 2014 | 59 |
| Fluorescence, temperature and narrow-band indices acquired from a UAV platform for water stress detection using a micro-hyperspectral imager and a thermal camera [44] | Zarco-Tejada PJ | Remote Sensing of Environment | 2012 | 59 |
| Evaluating Multispectral Images and Vegetation Indices for Precision Farming Applications from UAV Images [45] | Candiago S | Remote Sensing | 2015 | 57 |
| Lightweight unmanned aerial vehicles will revolutionize spatial ecology [46] | Anderson K | Frontiers in Ecology and the Evironment | 2013 | 56 |
| Estimating Biomass of Barley Using Crop Surface Models (CSMs) Derived from UAV-Based RGB Imaging [47] | Bendig J | Remote Sensing | 2014 | 55 |

Zarco—Tejada PJ used an UAV platform equipped with a micro-hyperspectral imager and a thermal camera for water stress detection [44]. Candiago S confirmed that high-resolution UAV data have great potential for precision agriculture assessments [45]. UAV RS technology research hotspots are in agricultural sciences and ecology, filling some short-comings in this field. For example, UAV-based RS offers great possibilities to acquire field data in a fast and easy way for precision agriculture applications [45]. Both multispectral and digital sensors mounted on UAVs are reliable platforms for rice growth and grain yield estimation. Using UAV RS can predict the best period and best vegetation index for assessing rice grain yield [48]. One of the advantages of UAV RS is high-resolution images. Some researchers use UAV micro-hyperspectral instrument configurations to perform fluorescence studies on trees, measuring their photochemical reflectance index and canopy temperature; stomatal conductance obtained from the instrument can be linked to field measurements of water potential [44].

Trend Analysis

Analyses of keywords and highly cited papers can help identify the key topics in a research field but may not be adequate for analyzing a field's most recent development trends. Both analyses ignore the changes that occur over time, and therefore do not retrieve the latest emerging literature. Burst detection is used to detect trends that appear rapidly across papers to predict the trends in their topics [17]. We have counted the top 25 references with the strongest citation bursts for at least three years in Figure 8. Each publication in the figure corresponds to a blue line. A darker shade of blue reflects the publication date of the paper and red reflects the period in which the paper was cited.

We divided the research period from 2010 to 2021 into three periods similar to those of keywords for the literature outbreak citation analysis to infer the future research trend of UAV RS. Combining the research content of the cited outbreak period and the relevant literature, it can be found that the initial UAV RS mainly used crop growth real-time monitoring and provided high-resolution images. Later, it gradually shifted its focus to improving the UAV visual effects. The leaf area index and photochemical reflectance images were further studied to provide more effective solutions for precision agriculture. In recent years, UAV RS has gradually transitioned to favor machine deep learning, neural networks, and RS image classification with new techniques in target recognition, image segmentation and other fields [37]. Therefore, we speculate that future UAV RS trends will be as follows. First, UAVs will develop more intelligently and will be applied to various systems. Second, UAV sensors will provide the advantages of strong versatility, reasonable cost, light weight, and small size. Sensor technology will be extensively studied and developed rapidly [49]. Third, because the number of images captured by UAV RS is huge and their processing is cumbersome, there will be research to develop quicker UAV image stitching approaches to cut down the subsequent processing time [50].

| References | Strength | Begin | End | 2010 - 2020 |
|---|---|---|---|---|
| Berni JAJ, 2009 | 15.45 | 2010 | 2014 | |
| Zarco-Tejada PJ , 2012 | 23.04 | 2012 | 2017 | |
| Hunt ER, 2010 | 12.96 | 2012 | 2015 | |
| Turner D, 2012 | 13.9 | 2013 | 2017 | |
| Watts AC, 2012 | 11.1 | 2013 | 2017 | |
| Wallace L, 2012 | 9.91 | 2013 | 2017 | |
| Zhang CH, 2012 | 19.14 | 2014 | 2017 | |
| Anderson K, 2013 | 11.67 | 2014 | 2018 | |
| Honkavaara E, 2013 | 10.2 | 2014 | 2018 | |
| Niethammer U, 2012 | 9.52 | 2014 | 2017 | |
| Baluja J, 2012 | 9.52 | 2014 | 2017 | |
| Harwin S, 2012 | 8.69 | 2014 | 2017 | |
| Koh LP, 2012 | 7.86 | 2014 | 2017 | |
| Kelcey J, 2012 | 7.86 | 2014 | 2017 | |
| Fonstad MA, 2013 | 7.27 | 2014 | 2018 | |
| dOleire-Oltmanns S, 2012 | 7.03 | 2014 | 2017 | |
| Torres-Sanchez J, 2013 | 6.73 | 2014 | 2017 | |
| Pena JM, 2013 | 6.64 | 2014 | 2018 | |
| Getzin S, 2012 | 6.62 | 2014 | 2017 | |
| Primicerio J, 2012 | 6.62 | 2014 | 2017 | |
| Westoby MJ, 2012 | 21.58 | 2015 | 2017 | |
| Mulla DJ, 2013 | 8.52 | 2015 | 2018 | |
| Dandois JP, 2013 | 8.52 | 2015 | 2018 | |
| Lisein J, 2013 | 7.51 | 2015 | 2018 | |
| Mancini F, 2013 | 8.65 | 2016 | 2018 | |

**Figure 8.** Top 25 papers published during the period 2010–2021 with the strongest citation bursts.

## 4. Discussion

### 4.1. Current Development of UAV RS Field

The number of papers published by China and the US was far ahead in the UAV RS field. It was worth noting that although Chinese institutions had a greater advantage in the number of publications, the centrality index of evaluating the influence of the paper was relatively low. This indicated that although Chinese scholars had considerable research team and ability in this field, the international comprehensive influence of the publications from China could be improved. China should strengthen the innovative research of publications in this field and enhance international influence in the future. The cooperation strength of the countries and institutions were relatively weak, indicating a scope for future improvement. The research community should pay more attention to the quality of papers and improve the research height of this field through cooperation between institutions.

The UAV RS field involved multi-disciplinary investigations. Trying to combine disciplines with a large span in the future will contribute to finding new research directions.

The UAV RS field can be applied to a wide range of topics. At first, people used simple images to monitor crop growth and rescue disaster. As the UAV visual effects and image manipulation techniques gradually improved, many new applications emerged. However, the data volume of UAV RS was enormous, so more attention should be paid to data processing. These developing areas of applications can greatly benefit from improved UAV RS data processing techniques. Moreover, upgrading the observed object from static to dynamic, and efforts finding the appropriate algorithm to accurately interpret the target were also considerable. Meanwhile, big data and artificial intelligence have developed rapidly, thus promoting the improvement of UAV RS relevant techniques and applications, such as neural networks for image segmentation. With machine learning technology advancements, UAV RS research has tended to become more intelligent. In addition, to equipping UAV with the functions of decisions making and responding to unexpected events in real time without direct human intervention, more technology and algorithms need to be developed.

### 4.2. Future Prospects of UAV RS Field

UAV RS is highly versatile, with applications in agriculture and forestry, ecology, and geology, among others. UAV RS applications are also growing in fields such as land resource survey, three-dimensional modeling, geographic national conditions monitoring, and neural network building [51,52]. We summarize future applications as follows.

First, UAV can be equipped with multiple sensors to achieve directional business services. For example, combining lidar and image modeling to obtain visible light images, not only provides texture characteristics for the model surface, but also encrypts and repairs the point cloud, which helps improve its reconstruction [53]. Second, stable and small UAV will be developed, UAV anti-jamming and long-distance transmission capabilities will reach higher standards [54]. UAV RS images can be transmitted in real-time, which is vital for military operations, emergency disaster response and emergency rescues [55]. Third, rapid stitching and image recognition tools will be developed. UAV RS can automatically check the quality of the flying photos and handle image feature points [56]. Fourth, 5G technology will be combined to realize the networking function control system, so that UAV can safely and efficiently enter the mid-to-high altitude airspace and expand the UAV networking scope [57]. Fifth, RS loads and professional operations will be more intelligently integrated, especially in the field of precision agriculture. For example, analyzing images of diseases and insect pests can help determine the exact location of infestations and implement precise spraying operations [58]. Finally, big data cloud processing platforms for UAV RS will be built. UAV RS data acquisition can be customized on-demand to meet the needs of phase, resolution, type, real-time for users [53,55].

### 4.3. Limitation of the Study

Some limitations in the analysis tools and the data screening process may still be affecting the accuracy and thoroughness of the results in this paper. First, the retrieved data may have missed some publications. We retrieved the data using "TS" as the search index, some publications that do not contain the searched keywords in their title, abstract, author keywords, and keyword plus may have been excluded from the records. Manual selection could also lead to the erroneous elimination of some publications. Additionally, some nodes in the original network could not be displayed in the figures. We also made some adjustment (e.g., we canceled or moved some nodes) to enhance the esthetics and clarity of the network. Finally, we thank the CiteSpace tool for providing the data visualization functions used in this paper and hope it will be improved in corresponding functions in the future.

## 5. Conclusions

In this study, we use the bibliometric analysis to conduct research of Unmanned Aerial Vehicle (UAV) Remote sensing (RS) related papers. The results indicated that the publication of UAV RS has shown an overall growing trend, particularly from 2015 to 2020. The publications from the United States (US) and China far exceeded those of other countries. China ranked first in the number of publications and was the only country with more than 500 publications. The US was the most active country in terms of international cooperation in the UAV RS field, exerting a wide and unrivaled range of influence. The publications of China were far more numerous than those of the US but with much less cooperation with other countries. South Korea ranked 12th in the number of publications, but its centrality value was zero, indicating that most of its research was done independently and with little cooperation with the outside world. The Chinese Academy of Sciences (CAS) was the top institution in the number of institution publications, indicating that the cooperation in the UAV RS field was concentrated rather than widely distributed. In addition, this domain involved multiple subjects from "Environmental Sciences and Ecology" to "Photographic Technology", "Agriculture", "Engineering and Electrical", "Computer Science" and "Physical Geography". Among them, "Environmental Sciences and Ecology" was the main related discipline. However, the lack of cooperation among countries and institutions will limit the development of UAV RS availability. Promoting cooperation among institutions and countries can push the continuous advancement of this field. UAV RS evolved from the direct observation of UAV RS images to the gradual improvement of image processing technology, and then to the development of machine learning technology, such as neural networks for image segmentation. Future research trends of UAV RS are expected to be related to artificial intelligence. Based on this, we briefly summarized the developments of the UAV RS applications over the past decades and raised some ideas about future applications in the discussion section. These conclusions are helpful to provide a scientific basis and guidance for the research community to better grasp the developments of the UAV RS field.

**Author Contributions:** Conceptualization, S.W. and J.W.; methodology, D.Z.; software, J.W.; validation, H.C., S.W. and R.Z.; formal analysis, J.W.; investigation, H.C. and H.L.; resources, K.Y. and W.Z.; data curation, J.W.; writing—original draft preparation, J.W.; writing—review and editing, D.Z., S.W. and H.L.; visualization, J.W.; supervision, S.W., W.Z. and K.Y.; project administration, S.W. and D.Z.; funding acquisition, K.Y. All authors have read and agreed to the published version of the manuscript.

**Funding:** This work was supported by the National Natural Science Foundation of China (41977415 and 41901298), the Fundamental Research Funds for the Central Universities (2652018031), and the open fund of Shanxi Key Laboratory of Resources, Environment and Disaster Monitoring (2019-04).

**Acknowledgments:** The authors would like to thank the "Quantitative Remote Sensing & Climate Change" (QRSCC) team of China University of Geosciences for their support and kind cooperation.

**Conflicts of Interest:** The authors declare no conflict of interest.

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
