# Peer review of "Social Network and Bibliometric Analysis of Unmanned Aerial Vehicle Remote Sensing Applications from 2010 to 2021"

_remotesensing, doi:10.3390/rs13152912_

Round 1
Reviewer 1 Report
This is an unusual but excellent paper for the remote sensing community. It covers the citation and social interaction of papers and terms/concepts within the field of remote sensing covering the past several decades, and shows the changes and directions of new citation and relevant status of different research papers and studies in this field. It is a well documented paper and scientifically sound, and should be well received by the community. It is almost an Applied Anthropology or Social-Network research paper which is important on its own showing the interaction and citation index relationships between different research groups and disciplines. In addition, it is very important to the current Remote Sensing research community especially taking into account the fact that direct communication between researchers has not been possible due to the pandemic these past years, so that social discourse which is often communicated within a normal conference setting has been replace by online interaction which is not as effective in showing the nuances of citation importance and self ranking of research groups and papers. Because of this, I feel it is an excellent Social Network paper in its own right, but is also important within the Remote Sensing Community.
Author Response
Thank you for providing us valuable suggestions and they do help improve the paper. For the specific comments , we have made detailed reply in file. We hope that the revision is acceptable.

Reviewer 2 Report
This is an interesting paper. I am a big fan of using SNA and bibliometrics to understand scientific cultures and changing fields, as you have done here. There are some sentences in the Introduction that need work. They are:
Line 31:
"Traditional satellite based Remote Sensing (RS) is limited by long return visits, poor timeliness, high information collection costs, and difficulty in capturing cloud-covered regions [1]." - You don't need to overstate challenges with traditional remote sensing to make your point. RS is not always limited by "long return times" and "poor timeliness". Plus, flying UAVs in clouds isnt always recommended either. Just state the case that RS is different: spatial res, control, choice of sensor, etc. No need to make RS a failure to advance your argument.
"Moreover, the temporal and spatial resolution of the acquired data is not suit- able for local applications [2]." Also not always true - Sentinel is often used for "local" applications. Just be straightforward about resolution.
Line 44: "Traditional review papers are not suitable for properly addressing the above questions". Again, you are making this an extreme case to advance your needs, but you do not have to! traditional review papers can do exactly what you say: they can "appreciate the key advances of this technology and how they have evolved." What you want to say is that traditional review papers look at the content of papers, you are using a novel method that looks at the characteristics of the authors, the institutions, and how the focus of the papers in this emerging field has evolved.
Line 89. Citation for software, and version.
Figure 1. Remove internal grid. It will make the map clearer. CHange "Amount of Publications" to "Number of Publications" in the caption.
Figure 2. CHange "Number of Literature" to "Number of Papers".
Line 373: Influence of China was "relatively insufficient" - maybe say it could be improved.
Line 437: "South Korea ranked 12th in the number of publications, but 437 most of its research was done independently and cooperate with the outside world little." Why call out SK when other countries are not similarly highlighted?
Author Response

(The authors gave the same response as above.)
